Subject Area:
developmental biology/neuroscience

Keywords:
extracellular matrix, development, tissue shape

Authors for correspondence:
Katherine R. Long
e-mail: long@mpi-cbg.de
Wieland B. Huttner
e-mail: huttner@mpi-cbg.de

# How the extracellular matrix shapes neural development

Katherine R. Long and Wieland B. Huttner

Max Planck Institute of Molecular Cell Biology and Genetics, Pfotenhauerstraße 108, D-01307 Dresden, Germany

 KRL, 0000-0003-0660-2486; WBH, 0000-0003-4143-7201

During development, both cells and tissues must acquire the correct shape to allow their proper function. This is especially relevant in the nervous system, where the shape of individual cell processes, such as the axons and dendrites, and the shape of entire tissues, such as the folding of the neocortex, are highly specialized. While many aspects of neural development have been uncovered, there are still several open questions concerning the mechanisms governing cell and tissue shape. In this review, we discuss the role of the extracellular matrix (ECM) in these processes. In particular, we consider how the ECM regulates cell shape, proliferation, differentiation and migration, and more recent work highlighting a key role of ECM in the morphogenesis of neural tissues.

## 1. Introduction

The development of the nervous system requires a complex and intricate balance between the proliferation of progenitors, differentiation of the correct cell types and the subsequent migration and connection of these cells. While significant progress has been made in understanding how each of these aspects is regulated individually, the exact mechanisms that govern their coordination remain largely unknown. However, this coordination is crucial to ensure the development of the correct shape and function of the nervous system. This is true on multiple scales, ranging from the shape of cell processes to the shape of entire tissues.

The importance of regulating tissue shape in neural development is highlighted by the many neurodevelopmental disorders that arise from defects in morphogenesis. During early neural development, these include neural tube defects such as anencephaly, a failure of upper neural tube closure that results in brain defects, and spina bifida, a failure of lower neural tube closure that results in spinal cord defects [1,2]. Later in development, these include disorders that lead to defects in the folding of the neocortex, such as lissencephaly, a reduction in cortical folding, and polymicrogyria, an increase in cortical folding, both of which can result in cognitive defects [3–5].

How tissue shape is regulated during neural development remains an unanswered question. A promising candidate for this is the extracellular matrix (ECM), the complex network of proteins that surrounds cells within a tissue [6]. ECM has long been known to not only be expressed within developing neural tissues in many different species but also to regulate many aspects of neural development [7–20]. Although it was once thought to act purely as a scaffold to support the surrounding cells, the ECM has since been found to have many more complex roles. In particular, it has recently been implicated in regulating the morphogenesis of the developing neural tube [21–23] and neocortex [24], directly impacting the shape of these developing neural tissues.

In this review, we will discuss how the ECM shapes neural development, focusing on how it regulates aspects such as cell proliferation, differentiation, migration and tissue morphology. We will not cover in detail the composition of the ECM in the developing nervous system, nor the details of neural development, which have expertly been reviewed elsewhere [11,13,25–32].

royalsocietypublishing.org/journal/rsob   Open Biol. **9**: 180216

# 2. Extracellular matrix and neural progenitor cells

ECM components and their receptors have been studied in neural progenitors for decades, and there are several key functions that appear to be conserved across many species. In particular, the regulation of progenitor proliferation, differentiation and effect on cell shape appear to be highly conserved. We will discuss how the ECM regulates these specific aspects of neural development, with a focus on the developing neural tube and neocortex.

## 2.1. Proliferation

During early neural development, the ECM and its related receptors have been shown to have many functions. They provide structural support, often in the form of a basement membrane, but also modulate or stimulate signalling pathways that drive proliferation, both directly and indirectly. The key families of ECM components and receptors that have been shown to regulate proliferation are the proteoglycans, laminins and integrins.

### 2.1.1. Proteoglycans

There are several major types of proteoglycans (reviewed in [33]) but in this section, we will focus on the heparan sulfate (HS) and chondroitin sulfate proteoglycans (CSPGs). Both consist of a core protein with one or multiple glycosaminoglycan chains attached [33,34], either HS or chondroitin sulfate (CS), respectively. These chains can be modified, for example, by the site of sulfation in CS chains [35], thought to affect the function of CSPGs. The CSPGs include the lectican family (brevican, neurocan, versican and aggrecan), phosphacan, CD44 and the transmembrane component NG2 [18,33]. The heparan sulfate proteoglycans (HSPGs) include the syndecans, the glypicans, agrin and perlecan.

Of the HSPGs, perlecan is an example of an ECM component that provides structural support and regulates the proliferation of neural progenitors. Disruption of perlecan by the addition of a functional blocking antibody was shown to cause disruption of the basement membrane in the chick neuroepithelium, resulting in cell detachment and protrusion into the lumen of the developing diencephalon [36]. In mouse neocortex development, disruption of perlecan resulted in basement membrane disruption that led to an over-expansion of the neuroepithelium [37]. This expansion was due to an over-migration of neurons past the disrupted basement membrane, leading to the presence of ectopic neurons in the meningeal layers and, in half of the embryos studied, exencephaly (malformation of neural tube formation resulting in extrusion of the brain from the skull) [37].

However, perlecan was also shown to have functions beyond structural support. In the mouse embryos that did not exhibit exencephaly, microcephaly (a smaller brain) was present instead [38]. Loss of perlecan caused a lengthening of the G1 phase of the cell cycle. This led to an overall reduction in progenitor proliferation, and therefore a consequent reduction in the number of neurons generated [38]. The loss of perlecan appeared to mediate these effects in mouse by reducing the spread of Sonic hedgehog [38]. This is consistent with findings in *Drosophila* neuroblasts [39], where mutations in the

*Drosophila* homologue of perlecan, *trol*, led to a reduction of both fibroblast growth factor (FGF) and Hedgehog (Hh) signalling. This subsequently led to a reduction of neuroblast proliferation, arresting the cell cycle in the G1 phase [39]. *Trol* was shown to mediate these effects by increasing the movement of Hh through the tissue, and by increasing the binding ability of Hh to its receptors [39]. Together, these data suggest that the functions of perlecan are highly conserved.

The regulation of neural progenitor proliferation appears to be a conserved function of many proteoglycans. Another HSPG, syndecan 4, has been shown to regulate proliferation of neuroepithelial cells within the developing zebrafish neural tube [40]. In contrast to perlecan, knockout of syndecan 4 resulted in an increase in proliferation, whereas over-expression leads to a reduction in proliferation [40]. Interestingly, syndecans are known to interact with and modulate the major family of ECM receptors, the integrins, [41,42], which are discussed in more detail in the following section.

Another family of HSPGs, the glypicans, has also been identified to regulate proliferation in the developing nervous system. Glypican 1 and 4 are expressed in the developing mouse neuroepithelium [43,44], and glypican 1 null mice were reported to have a decrease in brain size, due to an inhibition of FGF signalling [44]. FGF signalling is also modulated by glypican 4, which promotes proliferation in the developing mouse neural tube via FGF2 [43]. This relationship between glypicans and FGF signalling appears to be evolutionarily conserved, as glypican 4 has also been shown to modulate FGF signalling in the *Xenopus* embryo to regulate early forebrain patterning [45].

The CSPGs also play a role in regulating the proliferation of neural progenitors. Within the developing mouse neocortex, disruption of CSPGs via the addition of chondroitinase ABC (the enzyme that degrades CSPGs) resulted in a reduction in neural progenitor proliferation and subsequent generation of neurons [46]. In addition, this loss of CSPGs then increased the number of astrocytes generated, suggesting a switch in progenitor fate from the neuronal to glial lineage [46]. Similar results were also shown in neurosphere studies, where loss of CSPGs via the addition of chondroitinase ABC reduced the proliferation of mouse neurospheres, while, conversely, the addition of CSPGs stimulated the formation and proliferation of neurospheres via the epidermal growth factor (EGF) pathway [47].

Although some roles of the proteoglycans appeared to be conserved, there are cases where the function of a specific component differs between species. For example, in contrast to the above studies in mouse, loss of CSPGs in rat neurospheres via chondroitinase ABC increased both proliferation and neuronal differentiation [48]. Addition of chondroitinase ABC to these neurospheres caused a change in shape, resulting in adherence of cells *in vitro* and a reduction in sphere formation [48]. This function of CSPGs required additional ECM-related molecules, as the effects of chondroitinase ABC were blocked by the addition of echistatin, a disintegrin (a highly potent inhibitor of integrin $\beta$1 and $\beta$3, isolated from snake venom), suggesting that this function of CSPGs was mediated by the integrin pathway [48].

### 2.1.2. Laminins and integrins

Two major families of ECM-related molecules are the laminins and their receptors, the integrins. Each laminin chain is

made up of an alpha, beta and gamma subunit, and from the five alpha, four beta and three gamma subunits currently known, 15 distinct laminin chains have been identified [49]. These are now named after their composition (i.e. laminin-111 comprises the subunits alpha 1, beta 1 and gamma 1). Many of these laminin chains have been shown to be expressed throughout neural development. In the mouse, for example, almost all of the laminin chains are expressed during neural tube formation (E9.5 in mouse [50]), and high levels of laminin alpha 2 and 4 were detected in the ventricular zone of the developing neocortex (E14.5 [51]).

Laminins are known to have a major function within the basement membrane [52], but as with the proteoglycans, they have also been shown to regulate neural progenitor proliferation. Until relatively recently, the specific laminin chains were not available as recombinant proteins, resulting in earlier studies often referring to laminin in general. This general laminin is thought to have contained a mixture of different laminins, but was often predominantly laminin-111.

Many of the studies on the effect of laminin on neural progenitor proliferation were conducted using *in vitro* systems. These have shown that the addition of laminin enhanced the proliferation of mouse neuroepithelial cells [53]. It also increased the proliferation and survival of human neural stem cells (NSCs) [54,55], and promoted the differentiation of NSCs from human embryonic stem cells (ESCs) [56]. Laminins are thought to promote these responses in two ways. The first way is by modulating growth factors, such as enhancing FGF signalling [53]. This is also observed in the adult NSC niche that lines the lateral ventricles in the mouse and human brain, the subventricular zone. Here, adult NSCs proliferate near laminin-rich structures, called fractones, that capture FGF2 [57]. The second way is by directly signalling via their receptors, the integrins.

The integrins are the major family of ECM receptors [58] and are highly expressed in the developing nervous system. In the developing mouse neocortex, the major integrin subunits to be expressed are integrin beta 1 ($\beta$1) and its dimer partner integrin alpha 6 ($\alpha$6). Integrin $\beta$1 was shown to be expressed throughout the early neuroepithelium [59,60], while the dimer of integrin $\beta$1 and alpha 6 ($\alpha6\beta$1) was more highly expressed along the ventricular surface in the developing mouse neocortex (E14.5) [51,59]. Expression of integrin $\alpha6\beta$1 in neural development is highly conserved. Thus, integrin $\alpha6\beta$1 was also shown to be expressed in the early chick neuroepithelium [61] and to be a marker of NSCs in the developing human neocortex [55,62].

In the developing mouse neocortex, increased expression of integrin $\beta$1 resulted in increased generation of neurospheres, while, conversely, loss of integrin $\beta$1 resulted in the generation of smaller neurospheres [59]. This effect of integrin $\beta$1 was due to its regulation of MAPK signalling, with lower levels of integrin $\beta$1 resulting in a reduction in signalling [59]. Similar results were obtained from experiments using neurospheres and an integrin $\beta$1 conditional knockout. Loss of integrin $\beta$1 again resulted in a reduction of progenitor proliferation, resulting in the generation of smaller neurospheres [63]. Additionally, these neurospheres also showed a reduced response to EGF, FGF and nerve growth factor signalling [63]. These data indicated that integrins can regulate neural progenitor proliferation by mediating the activity of major signalling pathways.

Integrin $\alpha6\beta$1 is not the only integrin dimer to have such an effect on cortical progenitor proliferation. Activation of another integrin, $\alpha v\beta$3, in the developing mouse neocortex also resulted in an increase in progenitor proliferation [64] (figure 1$a-c,e$). Specifically, administration of activating antibodies to E14.5 mouse neocortex increased the proliferation of basal progenitors (BPs). This is particularly interesting, as the proliferative capacity of BPs is thought to be partly responsible for the evolutionary expansion of the neocortex. The $\alpha v\beta$3-mediated increase in proliferation was due to an increase in both cell cycle re-entry and in the number of symmetric proliferative divisions of BPs. Interestingly, these effects of integrin $\alpha v\beta$3 activation required the binding of thyroid hormones to integrin $\alpha v\beta$3, providing a possible explanation for the impairment of cortical development upon lack of thyroid hormones during pregnancy [64].

These data suggest that the regulation of cortical progenitor proliferation by integrins may have contributed to the expansion of the neocortex. Two studies in the developing chick and ferret provided further support for this. Compared with mouse, the ferret has an increased number of proliferative BPs, and therefore a more expanded and folded neocortex. Addition of echistatin, a disintegrin, to developing ferret neocortex slice cultures reduced the proliferation of these BPs [66], suggesting integrin signalling was required to maintain their high proliferative capacity. The second study was in the developing chick neuroepithelium. The expression of a constitutively active integrin $\beta$1 in this system led to an increase in the proliferation of the neuroepithelial cells [61]. This resulted in the generation of basally dividing progenitors (not usually found in the chick neuroepithelium), increased both the number of progenitors and neurons generated, and subsequently led to a radial expansion of the neuroepithelium [61]. These are all key characteristics of mammalian species with an expanded neocortex, suggesting that integrins may play an important role in regulating neocortical size.

### 2.1.3. Role of extracellular matrix in neocortex expansion

Many of the functions of ECM components and receptors on progenitor proliferation appear to be evolutionarily conserved. However, recent transcriptomic studies have indicated an increase in ECM expression in the developing human neocortex compared to the mouse [67–71]. This is especially true for those areas and progenitors with an increased proliferative potential in the human: the OSVZ and the BPs that reside within it. Together with the data showing a role of integrin signalling in BP expansion, these data suggest that the changes in ECM expression observed in the developing human neocortex may have contributed to its evolutionary expansion via the regulation of progenitor proliferation.

## 2.2. Differentiation

Various ECM components have also been shown to promote the differentiation of neural progenitors. Neuroepithelial cells in the developing chick retina proliferated and differentiated into neurons when plated on laminin-1, and expressed the specific laminin chain alpha 1 at the time of neuronal differentiation [72]. This specific chain appears to be required for differentiation, as plating of the neuroepithelial cells onto laminin lacking the alpha 1 chain only promoted proliferation of these cells [72]. This effect of laminin was also observed with early neural tube progenitors in the chick embryo.

royalsocietypublishing.org/journal/rsob    Open Biol. 9: 180216

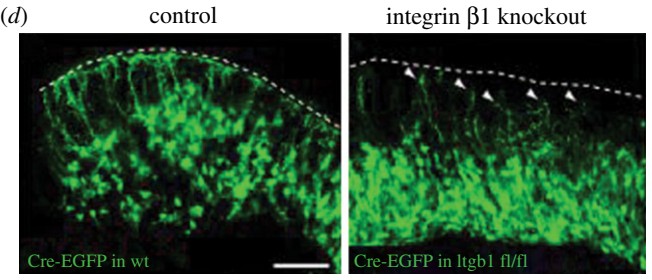

**Figure 1.** ECM, integrins and progenitor cell behaviour. (a) Images showing E14.5 mouse neocortex after 24 h of whole hemisphere (HERO) culture with either control IgG antibody (left panels) or the integrin αvβ3 activating antibody, LIBS-6 (right panels), with DAPI staining (blue, upper panels) and immunofluorescence for the mitotic marker PH3 (white, lower panels). White dashed lines delineate the ventricular zone (VZ) and subventricular zone (SVZ) boundary. Scale bar represents 50 μm. Adapted from [64]. (b,c) Quantification of PH3 positive (mitotic) cells in the (b) VZ (APs) and (c) SVZ (BPs). Adapted from [64]. (d) Images showing GFP in the neocortex of E11 wild-type (left) and Itgb1$^{flox/flox}$ (knockout) (right) mice that were electroporated with CRE-IRES-eGFP and analysed after 24 h. White dashed lines delineate the basal surface. White arrowheads delineate detached radial glia processes. Scale bar represents 100 μm. Adapted from [65]. (e) Schematic summarizing the effects of blocking, knocking out or activating ECM and integrins on neuroepithelial and radial glial cell behaviour.

Plating these cells onto laminin permitted neuronal differentiation, and cells preferentially extended neurites towards the laminin coating, as opposed to collagen [73]. These effects were blocked by addition of an anti-laminin antibody, suggesting that laminin was required for the promotion of neuronal differentiation [73]. This effect of laminin was not limited to chick neural progenitors, as plating both mouse and human NSCs onto laminin also promoted neuronal differentiation (mouse [53]; human [56]).

This differentiative effect of laminin appears to contradict its pro-proliferative effects, suggesting there is a more complex network of signals that maintain the balance between laminin-induced proliferation and differentiation. Within the developing neural tissue, this would allow cells in a similar ECM environment to have opposing behaviours. Such a complex network has been shown in the developing chick neuroepithelium. Expression of a constitutively active integrin β1 in the early neuroepithelium resulted in an increase in proliferation of integrin β1-expressing cells, but promoted the differentiation of neighbouring cells lacking the active integrin β1 [61]. This involved a complex network of ECM components and signalling factors. The active integrin β1 drove increased proliferation via focal adhesion kinase (FAK) signalling. These proliferating cells increased expression of Wnt7a, which when secreted reinforced proliferation of these cells and promoted the expression of the ECM component decorin in neighbouring cells. Secretion of

decorin then interacted with the TGFβ receptor on the non-expressing cells, promoting their differentiation into neurons [61]. Interestingly, decorin is not normally expressed in the chick neuroepithelium at this stage (it is expressed during neural tube formation [21]). This suggests that the tissue may have responded to the increased proliferation caused by integrin β1 activity by secreting decorin to promote differentiation, in an effort to restore the balance between the two.

Within this simple system, the ECM was able to drive proliferation and differentiation in neighbouring cells using a network of ECM components, signalling factors and receptors. In more complex mammalian systems, with more progenitor cell types and germinal zones, it is highly likely that the ECM uses an even further complex network, in coordination with signalling factors and receptors, to regulate progenitor behaviour. Notably, the response of any cell to a specific ECM component will be greatly affected by the other factors and receptors acting on it. Together, this suggests that the picture of ECM regulation of progenitor behaviour within neural tissues is far more intricate than initially imagined.

## 2.3. Cell shape

### 2.3.1. Progenitors

In addition to regulating proliferation and differentiation, ECM can also alter the shape of neural progenitors and

neurons. A common theme among many studies is the role of the basement membrane for structural support of the developing nervous system and maintaining cell shape [65,74]. This is evident in the developing neocortex, where attachment of the apical radial glia (aRG) endfeet to the basement membrane via integrin $\beta$1 is important for the bipolar shape of these progenitors. Knockout of integrin $\beta$1 in the developing mouse CNS resulted in the detachment of the aRG basal processes from the pial basement membrane in the neocortex, causing subsequent apoptosis of these progenitors [65] (figure 1d,e). This detachment was also caused by knockout of the two laminin chains alpha 2 and 4 from the basement membrane, and the removal of the meninges, which greatly reduced the level of laminin within the basement membrane [65].

A similar effect was also observed when an integrin $\beta$1 blocking antibody was injected into the ventricle of the developing mouse neocortex, resulting in detachment of the aRG apical process from the ventricular surface [60] (figure 1e). Loss of the laminin alpha 2 chain from these aRG also induced detachment of the apical process [60], suggesting that both integrin $\beta$1 and laminin alpha 2 are required to maintain attachment of both the apical and basal process of aRG in the developing mouse neocortex.

In both of these examples, attachment of the aRG processes to either the apical or basal ECM, via integrin $\beta$1, is required for aRG to maintain their characteristic bipolar shape. This raises the open question as to whether attachment of basal radial glia (bRG, a more basally located progenitor in the neocortex that lacks an apical process) to the basement membrane is also required to maintain their basal process and overall morphology.

### 2.3.2. Neurons

Later in neural development, ECM also modulates neuronal shape, such as neurite extension (reviewed in [7]). In particular, laminin has long been associated with promoting neurite outgrowth [75]; reviewed in [76]). More recent work showed that plating both mouse and human NSCs onto laminin resulted in an increase in neurite extension and an increase in the complexity of neurites formed, in comparison to fibronectin or Matrigel (a soluble basement membrane extract) [55]. This was also the case for differentiating neurons from human ESCs; neurite extension was again increased when plated on laminin compared to various other ECM substrates [56]. The effect of laminin was partially disrupted by the addition of an integrin $\alpha$6 or integrin $\beta$1 blocking antibody [56]. This function of laminin is further supported by knockout of the laminin chain gamma 1 in developing mouse cerebral cortex, which led to reduced neurite outgrowth and abnormal axon pathfinding in vivo [77].

This function of laminin in regulating neurite growth is also present in other neural tissues. Retinal neurons were able to prolong their ability to extend neurites when plated on laminin after the activation of both integrin $\alpha$6$\beta$1 and $\alpha$3$\beta$1 [78]. These data suggested that the activation state of integrins may be an important factor in responding to the ECM environment [78], and could potentially provide a way of the cell to regulate its own response to the ECM environment.

Laminin is not the only ECM component to regulate neurite growth. Blocking the hyaluronic acid (HA) receptor

RHAMM (receptor for HA-mediated motility, also known as CD168) with function-blocking antibodies resulted in a reduction in neurite movement and migration in both rat and human primary neurons in culture [79]. This study also highlighted that the ECM can not only aid in the regulation of cell shape but also of the movement of cells; a vital process in the development of connections within the brain.

## 3. Extracellular matrix and cell migration

The most well-known ECM component that regulates neuronal migration is the glycoprotein Reelin [80–82]. However, in addition to this, many other ECM components have also been shown to influence the migration of both progenitors and newborn neurons during cortical development.

### 3.1. Progenitors

A key characteristic of neuroepithelial cells and aRG is the movement of the nucleus and cell body during the cell cycle—a process called interkinetic nuclear migration (INM). During INM, the nucleus and cell body of the progenitor migrate towards the basal side of the neuroepithelium to undergo S-phase. They then migrate down to the apical, ventricular surface to undergo mitosis [83–86]. Laminins have been suggested to play a role in this process in the zebrafish neural tube. Mutation of tab, a gene encoding laminin gamma 1, causes disruption of INM, resulting in mitoses of neuroepithelial cells in basal locations [87] (figure 2a,d). This function of laminin gamma 1 required signalling via FAK, as a morpholino targeting FAK also resulted in basal mitoses. Strikingly, neither cell shape nor polarity was affected, although the mitotic cleavage plane became randomized [87]. This suggests that this disruption of INM was an effect on the movement of the cell, and was not caused by the disruption of cell attachment to the basal or apical surface.

Other ECM components have also been shown to effect progenitor migration earlier in development, regulating the migration of neural crest cells (reviewed in [9]). The HA receptor RHAMM was shown to be expressed by cranial neural crest cells in Xenopus [88] and quail [89]. In the developing quail embryo, high concentrations of HA increased the number of neural crest cells generated from neural tube explant cultures [89]. HA promoted this generation of neural crest cells by reducing the cell–cell contacts between them, allowing a looser organization of the cells and their migration away from the neural tube [89]. This role of HA in neural crest cell migration is consistent with the notion that the higher level of HA observed in development, compared to the adult brain, aids the migration of newborn neurons by increasing the water content within the developing brain [90].

Several proteoglycans have also been reported to regulate neural progenitor migration, in particular, the lecticans, the major family of soluble CSPGs within the developing CNS [18,25]. They include brevican, neurocan, versican and aggrecan, of which versican has been shown to inhibit neural crest cell migration in Xenopus embryos [91]. This is consistent with findings in the developing chick embryo that both versican and aggrecan regulate neural crest cell migration [92].

Neural crest cells have also been shown to modulate their ECM environment to affect their own migration. Enteric

royalsocietypublishing.org/journal/rsob Open Biol. 9: 180216

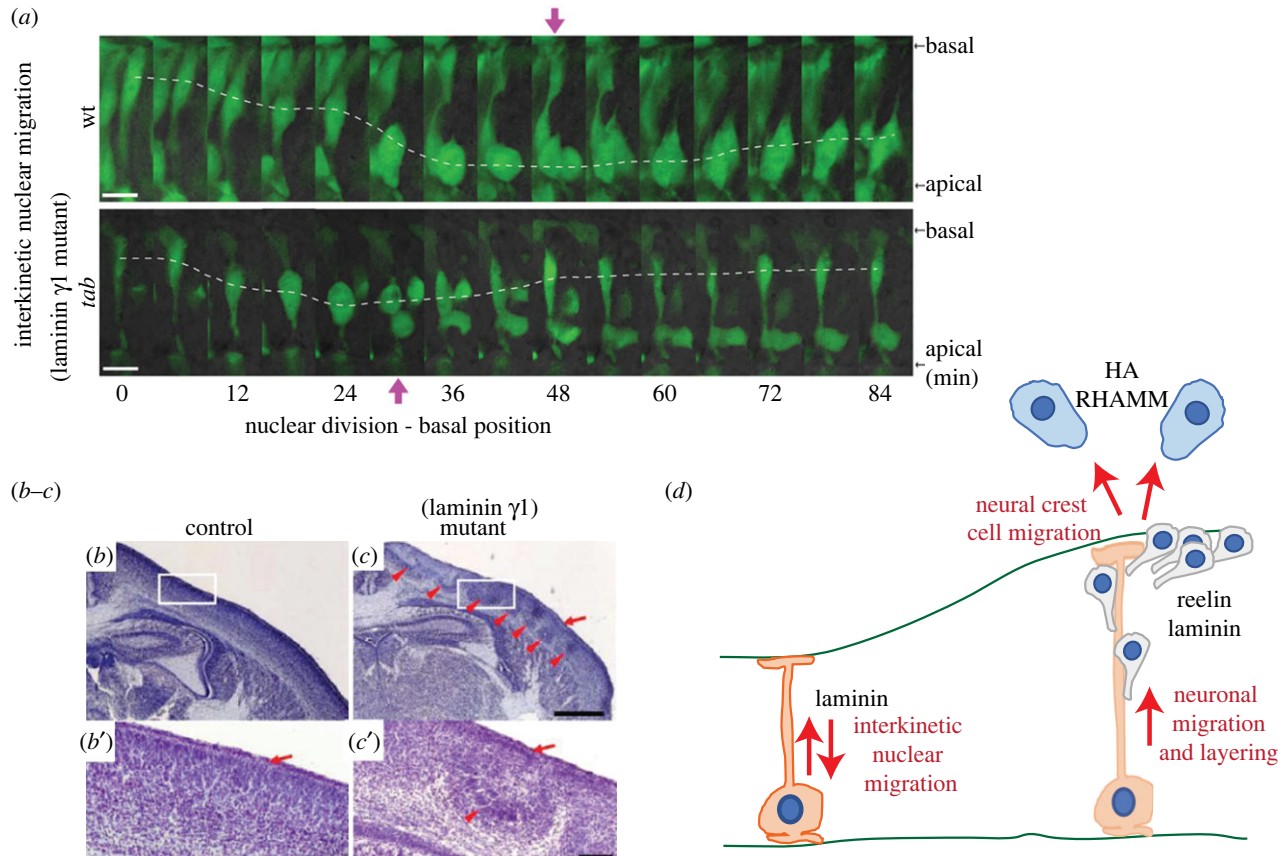

**Figure 2.** ECM and cell migration. (*a*) Images showing INM of GFP expressing neuroepithelial cells in the neural tube of the 9–10 somite stage wild-type (top panels) or *tab* mutant (lower panels) zebrafish. White dashed lines delineate the path travelled by the centre of the nucleus. Purple arrows indicate the nuclear division. Note the apical division of the nucleus in the wild-type, but the more basal division in the *tab* mutant. Scale bar represents 10 μm. Adapted from [87]. (*b,c*) Coronal sections of the neocortex of newborn P0 wild-type (*b,b′*) or laminin gamma 1 mutant (*c,c′*) mice. White boxes delineate areas shown in *b′* and *c′*. Red arrows indicate the marginal zone in the wild-type (*b′*) and abnormal organisation in the laminin gamma 1 mutant (*c,c′*). Scale bars represent 500 μm (*b,c*) and 50 μm (*b′,c′*). Adapted from [77]. (*d*) Schematic summarizing the effects of ECM on neural progenitor and neuronal migration.

neural crest cells (the cells that make up the enteric nervous system in the gastrointestinal tract) within the developing chick embryo were shown to initially express collagen XVIII, which promoted their migration [93]. However, later in development, towards the end of neural crest cell migration, these cells then switched to express the proteoglycan agrin, which inhibited neural crest cell migration [93].

## 3.2. Neurons

The classical example of ECM regulating neuronal migration is the role of Reelin in neocortex development. Reelin is secreted by Cajal-Retzius cells within the marginal zone of the developing neocortex [81,94], where its loss results in abnormal neuronal migration and defective cortical lamination [81]. Reelin is thought to mediate neuronal migration by promoting the branching of migrating neurons, allowing greater interaction with the ECM and radial glial scaffold [95]. Reelin has also been shown to be involved in the switch of neuronal migration from locomotion, when neurons migrate radially along radial glial processes up to the cortical plate, to terminal translocation, when neurons migrate quickly and independent of radial glial processes [96,97]. Reelin is thought to regulate these processes via the activation of integrin $\alpha5\beta1$, allowing the neurons to bind the local ECM ligand fibronectin [97]. Reelin has also been shown to regulate neuronal migration via the classical pathway; binding to the transmembrane receptors apolipoprotein E receptor 2 (ApoER2) and the very low-density

lipoprotein receptor (VLDLR), leading to phosphorylation of the downstream adaptor protein Disabled-1 [98].

The cortical delamination defects observed after the loss of reelin in mice are in-line with the effects of mutations in reelin (RELN) observed in human patients. First, loss of function mutations in RELN are associated with lissencephaly (a reduction in cortical folding) [99], and second, the persistence of reelin-expressing Cajal-Retzius cells was observed in patients with polymicrogyria (an excess of folding) [100]. These studies indicate that regulating not only the level of expression of reelin but also the timing of this expression, is vital for the normal migration of neurons, and consequently the shape and folding of the developing neocortex.

Although other ECM components have also been implicated in early neuronal migration, such as the proteoglycans [19], it is often difficult to distinguish between the function of these ECM components in regulating migration and in maintaining basement membrane integrity. For example, mutation of the nidogen binding site of laminin gamma 1 resulted in detachment of RG from the basement membrane and disruption of cortical plate lamination [74]. It was later found that loss of laminin gamma 1 in cortical neurons disrupted the migration of neurons up to the marginal zone (figure 2*b–d*) and caused abnormal axonal pathways [77], a direct effect on migration. However, mutations in perlecan also show disruption of cortical plate lamination [38], but this effect is probably caused by the changes in basement membrane structure and altered

royalsocietypublishing.org/journal/rsob   Open Biol. 9: 180216

proliferation observed (which are described above in §2.1). It is, therefore, difficult to distinguish if many of the ECM components studied have a direct or indirect effect on neuronal migration. In spite of this, it is clear that the correct formation of the surrounding ECM has a major impact on the migration of newborn neurons, either directly or indirectly, which leads to a subsequent effect on tissue shape.

# 4. Extracellular matrix and morphogenesis

The regulation of neural progenitor behaviour by the ECM can indirectly affect the shape of developing neural tissues. However, more recent studies have shown that the ECM can directly affect morphogenesis, in both early and late neural development.

## 4.1. Early neural development

Many of the studies of morphogenesis in early neural development have focused on the neural tube in non-mammalian systems, such as the developing chick, zebrafish and *Drosophila*. Within the developing chick embryo, the small leucine-rich proteoglycan decorin was shown to regulate neural tube development [21]. Addition of a decorin neutralizing antibody to the embryonic neural tube resulted in disorganization of the neuroepithelium and disruption of neural tube closure [21]. In addition to this, the neural crest cells failed to migrate correctly, remaining above the neural tube [21]. This suggested that decorin plays a role in regulating both cell polarization and tissue morphogenesis during chick neural tube formation.

In the developing zebrafish, laminin and fibronectin have also been shown to regulate tissue movements during neural tube formation. During neurulation, the developing neural plate and mesoderm move in a coordinated manner, and are connected by laminin and fibronectin [22]. Depletion of both laminin gamma 1 and fibronectin resulted in the neural plate and mesoderm layers moving independently of each other [22], suggesting that both ECM components were required to couple the movements of these sheets of cells. This was specific to laminin gamma 1 and fibronectin, as depletion of HA, expressed only by the mesoderm, slowed mesoderm migration, but did not interrupt the coupled movements of the mesoderm and neural plate [22]. Combined with the data on decorin in the chick, this suggests that the ECM can coordinate the movement of cells and tissues in key morphogenetic events in neural tube development.

In order for ECM to play a role in tissue morphogenesis, it must also be modulated to allow cell and tissue movements to occur. Evidence for this is found in the *Drosophila* nerve cord. During development, the surrounding glia secrete the ECM that forms the neural lamella around the nerve cord. Disruption of this lamella, by expression of the metalloproteases MMP1 and 2, resulted in an abnormal, elongated shape of the developing nerve cord [101]. Regulation of CNS shape appears to be a conserved function of the metalloproteases. A mutation that reduced the function of the metalloprotease ADAMTS-A, the *runaway* mutant, resulted in migration of neural progenitors out of the developing nerve cord, dramatically altering its shape [23] (figure 3*a,c*). ADAMTS-A is suggested to regulate CNS shape by maintaining the structural integrity of the basement membrane

surrounding the nerve cord. One way in which it does this is by reducing the level of collagen IV present, which in turn reduces the stiffness of the ECM. The abnormal CNS shape caused by the reduced function of ADAMTS-A could therefore be rescued by depleting collagen IV, or by depleting its receptor, beta PS-integrin [23]. Together, these data indicate that regulating the stiffness of the ECM may be important for correct tissue morphogenesis [23].

The majority of the studies on early neural tissue morphogenesis have been conducted using the models outlined above. However, the recent development of various neural organoid systems has allowed some of these questions to be addressed in human tissue [102,103]. One such recent example is the function of laminin gamma 1 in human retinal organoids [104]. Blocking laminin gamma 1 function in these organoids, with a functional-blocking antibody, resulted in abnormal lamination and organization of the developing retina, as well a later degeneration of the shape of the retinal organoids [104]. Owing to the recent advances in the organoid field, it is becoming easier to study such morphogenesis events in early human neural development [103,105,106], and, given time, we predict further roles for the ECM in these early morphogenetic events of human neural development will be discovered.

## 4.2. Late neural development

Key morphogenetic aspects of late neural development have been recently shown to also be regulated by ECM, such as neocortex size and folding. Several studies have shown that altering either the ECM, or the ECM receptors integrins, caused changes to overall cortex morphology and shape [107]. Loss of integrin $\beta$1 in the developing mouse neocortex caused microcephaly [65] and disruption of integrin $\alpha$6 resulted in abnormal cortical plate lamination [74,108,109]. Additionally, mice with various laminin mutations display a disrupted organization of the basement membrane, an abnormal lamination of the cortical plate, and the presence of neuronal ectopias above the basement membrane [74,77,110,111]. However, in many of these models, it is difficult to identify if the ECM-induced changes in morphogenesis are caused by a direct effect on tissue shape, or by abnormal progenitor proliferation, neuronal migration or basement membrane structure.

More recently, ECM has been shown to directly alter the morphology of the developing human neocortex. The specific ECM components HAPLN1, lumican and collagen I were shown to induce folding of the cortical plate in human fetal neocortex explant cultures [24] (figure 3*b,c*). This folding was induced by the addition of recombinant HAPLN1, recombinant lumican and purified collagen I to these neocortex explants cultures as soluble components, and required this specific combination of all three ECM components. Strikingly, this ECM-induced folding occurred within 24 h of culture of 11–16 gestation week (GW) human neocortical tissue, an age prior to the onset of physiological folding, and involved a complex network of factors. These included an increase in the endogenous levels of HA within the tissue, local changes in ECM stiffness, and the HA receptor RHAMM (also known as CD168) and its downstream ERK signalling. HA was a key component of this folding, as depleting HA from the human neocortical tissue cultures could both block and reverse the ECM-induced

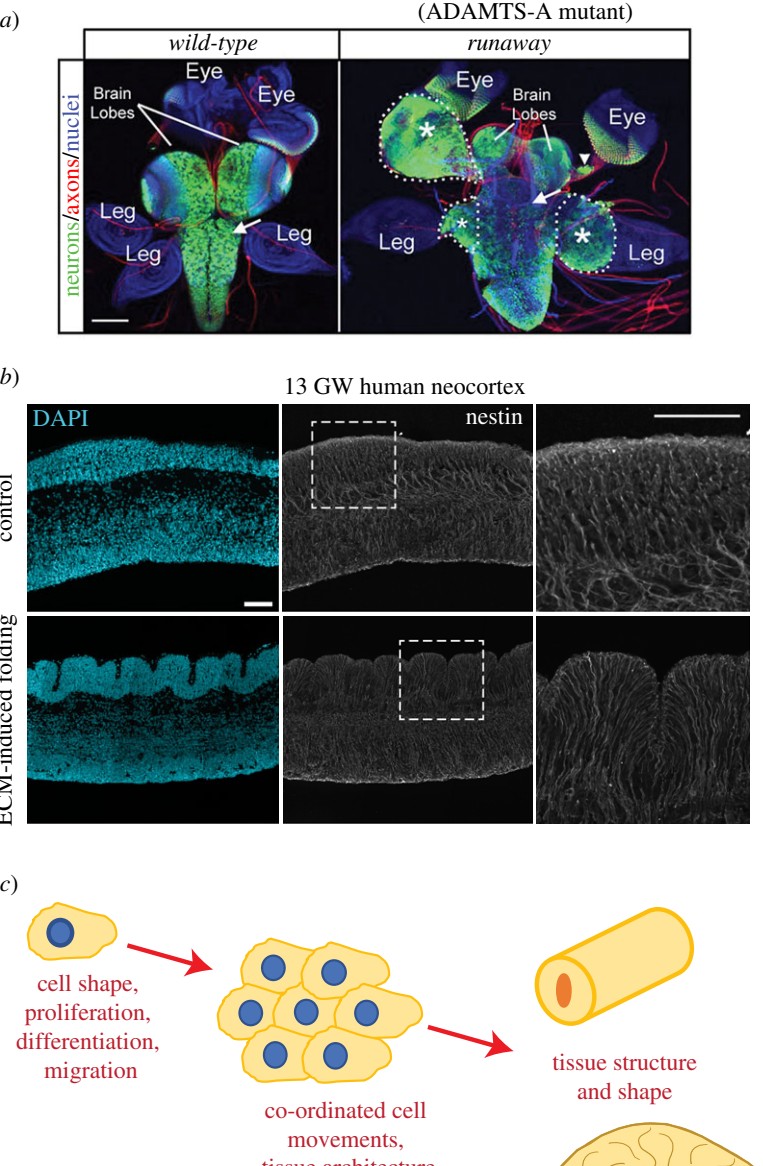

**Figure 3.** ECM and morphogenesis. (*a*) Images showing the ventral view of the CNS of late-third instar *Drosophila* larvae from the wild-type (left panel) and *runaway* mutant (right panel), with DAPI staining (blue), immunofluorescence for the neuronal marker ELAV (green) and axonal marker HRP (red). White dashed lines delineate the neuronal structures, which are also marked by white asterisks. Arrows indicate loss of neurons, arrowhead indicates small neuronal cluster. Scale bar represents 100 µm. Adapted from [23]. (*b*) Images showing 13 gestation week (GW) human fetal neocortex after 24 h of culture in control (upper panels) or after the addition of ECM components HAPLN1, lumican and collagen I, which induce folding of the cortical plate (lower panels), with DAPI staining (blue) and immunofluorescence for the radial glial process marker nestin (grey). White dashed boxes delineate areas shown in the panels on the right. Scale bar represents 50 µm. Adapted from [24]. (*c*) Schematic summarizing the effects of ECM on neural morphogenesis at the cellular and tissue levels.

folding [24]. These data suggest that modulating the ECM can greatly affect the morphology of the developing neocortex. In further support of this, depleting HA from later stage human fetal neocortex tissue, at 22 GW—when physiological folding has begun—reduced the level of folding inherently present, indicating a physiological role of HA in maintaining folding in the cortical plate of the developing human neocortex [24]. In the light of the important role of HA, it would be interesting to investigate the role of the HA-binding lecticans in cortical folding, especially as the transcripts of all four— aggrecan, brevican, neurocan and versican—are expressed in the developing human neocortex [68].

Taken together, these data strongly indicate that the ECM plays a key role in regulating the morphogenesis of the developing nervous system (figure 3*c*). With the ever-increasing development of new tools and model systems, such as cerebral organoids, it is highly likely that further roles for the ECM in these processes will be discovered.

## 5. Future perspectives

The role of the ECM in shaping the developing nervous system appears to be highly complex. The same ECM component can have multiple, even opposing, roles during neural development and can affect neighbouring cells in different ways. The ECM can also affect shape on a variety of scales, from individual neuronal processes to entire tissues.

royalsocietypublishing.org/journal/rsob   Open Biol. 9: 180216

Despite the many advances in recent years, it is clear that we are yet to understand the full complexity of the functions of the ECM. However, newly developed culture systems, tools and techniques are providing promising insight.

## 5.1. Stiffness

Several recent papers have started to probe how the stiffness of tissue and ECM can alter neural development. These include atomic force microscopy (AFM) measurements of stiffness in the developing mouse neocortex [112], the ECM of the developing human neocortex [24], and of *Xenopus* retinal ganglion cell axons [113]. The latter study showed that these axons preferentially grow towards an environment that is less stiff. Reducing the ECM stiffness *in vivo*, by addition of CSPGs, caused axons to migrate towards this area and become more disperse [113]. By contrast, when the environment was made stiffer, by applying a constant pressure using the AFM probe, the axons then avoided this area [113].

## 5.2. Hydrogels

Another promising new tool in which to study the functions of the ECM are hydrogel systems. Hydrogels have been used to generate a three-dimensional matrix for the generation of neural tube organoids, creating a reproducible and accessible system in which to study morphogenetic events in early mouse and human neural development [114]. They also allow for a more controlled environment, as ECM can be tethered to the gel or even printed into nanotopographic cues [115,116]. An example of the former showed that a HA-modified hydrogel, with RGD ligands, could promote the outgrowth of neurites in three dimensions from hippocampal neural progenitor cells [116]. This outgrowth was effected by the stiffness of the hydrogel, with greater outgrowth observed at a lower stiffness [116].

## 5.3. Nanotopographic extracellular matrix patterns

Hydrogel systems have also been used for more intricate experiments using nanotopographic printed lines, squares and grids of ECM components, such as laminin, fibronectin and collagen [115]. When NSCs were plated onto these nanotopographic ECM cues, more neurons were generated on the grid of laminin, whereas more astrocytes were generated on the lines and squares of laminin [115]. These data indicated that not only was the composition of the ECM present able to direct differentiation but also the shape and pattern of the ECM deposited. Additionally, nanotopographic printed

laminin lines could also orientate neurite outgrowth and promote neurite extension [117]. More recently, it was shown that ECM can also be printed in a nanotopographic gradient [118], opening up further lines of research to investigate how density and patterns of ECM components can direct neural cell shape and behaviour.

## 5.4. Extracellular matrix and neocortex expansion

The density and pattern of ECM composition becomes an interesting question for neocortex development and expansion. ECM components were reported to be more highly expressed in the developing human neocortex compared to the mouse [67]. If the density, pattern and gradient of ECM components can direct progenitor and neuronal cell behaviour, could this play a role in the development and expansion of the human neocortex? For example, the ECM microenvironment in the germinal zones could direct progenitor cell fate, direct neuronal migration, or aid the tangential migration of neurons observed in species with a folded neocortex [110,119–121].

## 5.5. Extracellular matrix biophysics

Finally, these functions of the ECM may be highly fluid and plastic, as the ECM itself is not a static structure, but rather mobile and dynamic [122]. Adding further complexity, cells can alter their own ECM environment very quickly, either by changing the expression of ECM components or by producing enzymes that degrade the ECM. This raises the question of how the forces generated by a mobile ECM can affect morphogenesis of the nervous system. Several possibilities are via signalling through integrins and cell–cell adhesions, forces generated by pushing and pulling of cells and the ECM, and by the constraint and promotion of cell and tissue movements (reviewed in [123,124]). With the further development of these tools, and the generation of new ones, these functions of the ECM will start to be uncovered, furthering our understanding of how the ECM shapes the developing nervous system.

Data accessibility. This article has no additional data.

Competing interests. We declare we have no competing interests.

Funding. W.B.H. was supported by grants from the DFG (SFB 655, A2), the ERC (250197) and ERA-NET NEURON (MicroKin).

Acknowledgements. While we have tried to include all the relevant work within the field, we apologize to anyone we have missed.

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
