## [Reviewer comments · Open Biology]

Review History

RSOB-18-0216.R0 (Original submission)

Review form: Reviewer 1

Recommendation

Accept with minor revision (please list in comments)

Are each of the following suitable for general readers?

- a) **Title**
Yes
- b) **Summary**
Yes
- c) **Introduction**
Yes

Is the length of the paper justified?

Yes

Should the paper be seen by a specialist statistical reviewer?

No

Is it clear how to make all supporting data available?

Not Applicable

Is the supplementary material necessary; and if so is it adequate and clear?

Not Applicable

Do you have any ethical concerns with this paper?

No

Comments to the Author

1) The major emphasis is on the work of the authors and hence on the basal lamina and interactions of the neural stem cells with these. Chondroitin sulfate proteoglycans are mentioned, but solely with reference to chondroitinase ABC treatment. Some information relating to the types of chondroitin sulfate and the families of core proteins that expose them should be provided.

2) The authors also mention heparin sulfate proteoglycans. They might mention that both syndecans and glypicans have been distinguished in neural tissues. Some of these proteoglycans regulate integrin functions, which is one focus of the review.

3) The authors discuss laminins and also their potential roles in FGF-signalling towards neural stem cells. Therefore, they should mention the fractones, laminin-based structures that have been proposed to serve as storage scaffolds for FGF2.

4) The authors devote some attention to reelin; therefore, they should also briefly comment on the classical signalling pathway.

5) In figure 3 the authors show the formation of gyri in an explant of human neocortex. They should specify how the matrix components were applied to the tissue – as soluble proteins or as plasmids/viruses? While details are part of the publication cited some information should be summarized here to obtain an understanding of experiment that is documented.

6) Because hyaluronic acid seems important in controlling neural folding the human cortex the authors might comment the expression of lecticans in the model they discuss.

7) In figure 1 E the presentation could be improved. While in the panels A-D it is indicated whether Itgs are blocked or stimulated this information is missing in the graphic scheme. Thereby it lacks clarity, it is not apparent whether stimulation or deletion/inactivation of the ECM components or receptors causes the effects regarding stem cell proliferation.

Decision letter (RSOB-18-0216.R0)

27-Nov-2018

Dear Dr Long,

We are pleased to inform you that your manuscript RSOB-18-0216 entitled "How the extracellular matrix shapes neural development" has been accepted by the Editor for publication in Open Biology. The reviewer has recommended publication, but also suggest some minor revisions to your manuscript. Therefore, we invite you to respond to the reviewer's comments and revise your manuscript.

Please submit the revised version of your manuscript within 14 days. If you do not think you will be able to meet this date please let us know immediately and we can extend this deadline for you.

- 1) A text file of the manuscript (doc, txt, rtf or tex), including the references, tables (including captions) and figure captions. Please remove any tracked changes from the text before submission. PDF files are not an accepted format for the "Main Document".
- 2) A separate electronic file of each figure (tiff, EPS or print-quality PDF preferred). The format should be produced directly from original creation package, or original software format. Please note that PowerPoint files are not accepted.
- 3) Electronic supplementary material: this should be contained in a separate file from the main text and meet our ESM criteria (see <http://royalsocietypublishing.org/instructions-authors#question5>). All supplementary materials accompanying an accepted article will be treated as in their final form. They will be published alongside the paper on the journal website and posted on the online figshare repository. Files on figshare will be made available approximately one week before the accompanying article so that the supplementary material can be attributed a unique DOI.

Online supplementary material will also carry the title and description provided during submission, so please ensure these are accurate and informative. Note that the Royal Society will not edit or typeset supplementary material and it will be hosted as provided. Please ensure that

the supplementary material includes the paper details (authors, title, journal name, article DOI). Your article DOI will be 10.1098/rsob.2016[*last 4 digits of e.g. 10.1098/rsob.20160049*].

4) A media summary: a short non-technical summary (up to 100 words) of the key findings/importance of your manuscript. Please try to write in simple English, avoid jargon, explain the importance of the topic, outline the main implications and describe why this topic is newsworthy.

Images

Data-Sharing

It is a condition of publication that data supporting your paper are made available. Data should be made available either in the electronic supplementary material or through an appropriate repository. Details of how to access data should be included in your paper. Please see <http://royalsocietypublishing.org/site/authors/policy.xhtml#question6> for more details.

Data accessibility section

Sincerely,

The Open Biology Team
<mailto:openbiology@royalsociety.org>

Reviewer's Comments to Author:

1) The major emphasis is on the work of the authors and hence on the basal lamina and interactions of the neural stem cells with these. Chondroitin sulfate proteoglycans are mentioned, but solely with reference to chondroitinase ABC treatment. Some information relating to the types of chondroitin sulfate and the families of core proteins that expose them should be provided.

2) The authors also mention heparin sulfate proteoglycans. They might mention that both syndecans and glypicans have been distinguished in neural tissues. Some of these proteoglycans regulate integrin functions, which is one focus of the review.

3) The authors discuss laminins and also their potential roles in FGF-signalling towards neural stem cells. Therefore, they should mention the fractones, laminin-based structures that have been proposed to serve as storage scaffolds for FGF2.

4) The authors devote some attention to reelin; therefore, they should also briefly comment on the classical signalling pathway.

5) In figure 3 the authors show the formation of gyri in an explant of human neocortex. They should specify how the matrix components were applied to the tissue – as soluble proteins or as plasmids/viruses? While details are part of the publication cited some information should be summarized here to obtain an understanding of experiment that is documented.

6) Because hyaluronic acid seems important in controlling neural folding the human cortex the authors might comment the expression of lecticans in the model they discuss.

7) In figure 1 E the presentation could be improved. While in the panels A-D it is indicated whether Itgs are blocked or stimulated this information is missing in the graphic scheme. Thereby it lacks clarity, it is not apparent whether stimulation or deletion/inactivation of the ECM components or receptors causes the effects regarding stem cell proliferation.

Author's Response to Decision Letter for (RSOB-18-0216.R0)

See Appendix A.

Decision letter (RSOB-18-0216.R1)

11-Dec-2018

Dear Dr Long

We are pleased to inform you that your manuscript entitled "How the extracellular matrix shapes neural development" has been accepted by the Editor for publication in Open Biology.

Sincerely,

The Open Biology Team
mailto: openbiology@royalsociety.org

Appendix A

Response to Reviewer's Comments

Reviewer's Comment:

1) *The major emphasis is on the work of the authors and hence on the basal lamina and interactions of the neural stem cells with these. Chondroitin sulfate proteoglycans are mentioned, but solely with reference to chondroitinase ABC treatment. Some information relating to the types of chondroitin sulfate and the families of core proteins that expose them should be provided.*

Authors' Response:

We have now included some brief information on the types of chondroitin sulfate and families of core proteins associated with them in section 2.1.1 of our review, and have also provided references that give a more detailed description of CSPGs, as follows.

“There are several major types of proteoglycans (reviewed in (Iozzo & Schaefer 2015)) but in this section we will focus on the heparan sulfate and chondroitin sulfate proteoglycans. Both consist of a core protein with one or multiple glycosaminoglycan chains attached (Prydz & Dalen 2000; Iozzo & Schaefer 2015), either heparan sulfate (HS) or chondroitin sulfate (CS) respectively. These chains can be modified, for example by the site of sulfation in CS chains (Purushothaman et al. 2012), thought to effect the function of chondroitin sulfate proteoglycans (CSPGs). The CSPGs include the lectican family (brevican, neurocan, versican and aggrecan), phosphacan, CD44 and the transmembrane component NG2 (Faissner & Reinhard 2015; Iozzo & Schaefer 2015). The heparan sulfate proteoglycans (HSPGs) include the syndecans, the glypicans, agrin and perlecan.”

Reviewer's Comment:

2) *The authors also mention heparin sulfate proteoglycans. They might mention that both syndecans and glypicans have been distinguished in neural tissues. Some of these proteoglycans regulate integrin functions, which is one focus of the review.*

Authors' Response:

We thank the reviewer for this helpful suggestion. We have now included a brief description of the functions of both syndecans and glypicans in neural progenitor proliferation in section 2.1.1 and neural progenitor migration in section 3.1 of our review, as follows.

Section 2.1.1.

“The regulation of neural progenitor proliferation appears to be a conserved function of many proteoglycans. Another HSPG, syndecan 4, has been shown to regulate proliferation of neuroepithelial cells within the developing zebrafish neural tube (Luo et al. 2016). In contrast to perlecan, knockout of syndecan 4 resulted in an increase in proliferation, whereas over-expression lead to a reduction in proliferation (Luo et al. 2016). Interestingly, syndecans are known to interact with and modulate the major family of ECM receptors, the integrins, (Roper et al. 2012; Morgan et al. 2007), which are discussed in more detail in the following section.

Another family of HSPGs, the glypicans, has also been identified to regulate proliferation in the developing nervous system. Glypican 1 and 4 are expressed in the developing mouse neuroepithelium (Ford-Perriss et al. 2003; Jen et al. 2009), and glypican 1 null mice were reported to have a decrease in brain size, due to an inhibition of FGF signalling (Jen et al. 2009). FGF signalling is also modulated by glypican 4, which promotes proliferation in the developing mouse neural tube via FGF2 (Ford-Perriss et al. 2003). This relationship between glypicans and FGF signalling appears to be evolutionarily conserved, as glypican 4 has also been shown to modulate FGF signalling in the Xenopus embryo to regulate early forebrain patterning (Galli et al. 2003).”

Section 3.1:

“Several proteoglycans have also been reported to regulate neural progenitor migration, in particular the lecticans, the major family of soluble CSPGs within the developing CNS (Bandtlow & Zimmermann 2000; Faissner & Reinhard 2015). They include brevican, neurocan, versican and aggrecan, of which versican has been shown to inhibit neural crest cell migration in Xenopus embryos (Szabó et al. 2016). This is consistent with findings in the developing chick embryo that both versican and aggrecan regulate neural crest cell migration (Perissinotto et al. 2000).”

Reviewer's Comment:

3) The authors discuss laminins and also their potential roles in FGF-signalling towards neural stem cells. Therefore, they should mention the fractones, laminin-based structures that have been proposed to serve as storage scaffolds for FGF2.

Authors' Response:

We have now added a brief description of laminin-rich fractones and their interaction with FGF and neural stem cell proliferation to section 2.1.2 of our review, as follows.

“This is also observed in the adult neural stem cell niche that lines the lateral ventricles in the mouse and human brain, the subventricular zone. Here, adult

neural stem cells proliferate near laminin-rich structures, called fractones, that capture FGF2 (Kerever et al. 2007)."

Reviewer's Comment:

4) The authors devote some attention to reelin; therefore, they should also briefly comment on the classical signalling pathway.

Authors' Response:

We have now included a brief summary of the classical reelin signalling pathway in section 3.2 of our review, as follows.

"Reelin has also been shown to regulate neuronal migration via the classical pathway; binding to the transmembrane receptors apolipoprotein E receptor 2 (ApoER2) and the very low-density lipoprotein receptor (VLDLR), leading to phosphorylation of the downstream adaptor protein Disabled-1 (D'Arcangelo et al. 1999)."

Reviewer's Comment:

5) In figure 3 the authors show the formation of gyri in an explant of human neocortex. They should specify how the matrix components were applied to the tissue – as soluble proteins or as plasmids/viruses? While details are part of the publication cited some information should be summarized here to obtain an understanding of experiment that is documented.

Authors' Response:

We have now included this information in section 4.2 of our review, as follows.

"This folding was induced by the addition of recombinant HAPLN1, recombinant lumican and purified collagen I to these neocortex explants cultures as soluble components, and required this specific combination of all three ECM components."

Reviewer's Comment:

6) Because hyaluronic acid seems important in controlling neural folding the human cortex the authors might comment the expression of lecticans in the model they discuss.

Authors' Response:

Unfortunately, there is currently only very limited information available on the expression of the lectican family in the developing human neocortex at the age we examined in our model. However, we have now included the information of the transcripts of this family of ECM components we obtained from our previously published developing human neocortex dataset (Florio et al, 2015; Science) in section 4.2 of our review, as follows. We agree with the

reviewer that it would be an interesting family of ECM components to investigate in this model.

“In light of the important role of HA, it would be interesting to investigate the role of the HA-binding lecticans in cortical folding, especially as the transcripts of all four; aggrecan, brevican, neurocan and versican, are expressed in the developing human neocortex (Florio et al. 2015).”

Reviewer’s Comment:

7) In figure 1 E the presentation could be improved. While in the panels A-D it is indicated whether Itgs are blocked or stimulated this information is missing in the graphic scheme. Thereby it lacks clarity, it is not apparent whether stimulation or deletion/inactivation of the ECM components or receptors causes the effects regarding stem cell proliferation.

Authors’ Response:

We have now included this information in Figure 1E.